# Treatment Trends and Combined Methods in Removing Pharmaceuticals and Personal Care Products from Wastewater—A Review

**DOI:** 10.3390/membranes13020158

**Published:** 2023-01-27

**Authors:** Paripurnanda Loganathan, Saravanamuthu Vigneswaran, Jaya Kandasamy, Agnieszka Katarzyna Cuprys, Zakhar Maletskyi, Harsha Ratnaweera

**Affiliations:** 1Faculty of Engineering, University of Technology Sydney (UTS), P.O. Box 123, Broadway, NSW 2007, Australia; 2Faculty of Sciences and Technology (RealTek), Norwegian University of Life Sciences, P.O. Box 5003, NO-1432 Ås, Norway

**Keywords:** pharmaceutical and personnel care products, advanced oxidation, membrane separation, adsorption, biological degradation

## Abstract

When discharged into wastewater, pharmaceuticals and personal care products (PPCPs) become microorganic contaminants and are among the largest groups of emerging pollutants. Human, animal, and aquatic organisms’ exposures to PPCPs have linked them to an array of carcinogenic, mutagenic, and reproductive toxicity risks. For this reason, various methods are being implemented to remove them from water bodies. This report critically reviews these methods and suggests improvements to removal strategies. Biological, physical, and chemical methods such as biological degradation, adsorption, membrane filtration, and advanced electrical and chemical oxidation are the common methods used. However, these processes were not integrated into most studies to take advantage of the different mechanisms specific to each process and are synergistic in the removal of the PPCPs that differ in their physical and chemical characteristics (charge, molecular weight, hydrophobicity, hydrogen bonding, structure). In the review articles published to date, very little information is available on the use of such integrated methods for removing PPCPs. This report attempts to fill this gap with our knowledge.

## 1. Introduction

There is a rising concern associated with emerging pollutants (EPs) that are dangerous to the environment and potentially seriously affect human health [1,2,3]. EPs are made up of a range of chemicals and compounds that either occur naturally or are more usually manufactured for various medical and other human needs or uses. With the world’s chemical turnover valued at EUR 3,475 billion in 2017, an increase in new chemicals entering the market and a growing volume of production can be expected [4]. Pharmaceuticals are synthetic micropollutants occurring in the aquatic environment above a potential natural background level but with concentrations remaining at trace levels, up to the microgram per litre range [5]. Pharmaceuticals are consumed by humans and animals for medical treatment and include antibiotics, anti-inflammatory drugs, hormones, etc. Personal care products (PCPs) are applied for medical, hygiene and cosmetic purposes and include disinfectants, fragrances, insect repellents, etc. [3]. Pharmaceuticals and personal care products (PPCPs), when discharged into wastewater, become microorganic contaminants and are among the largest groups of EPs [6,7]. Contemporary wastewater treatment plants (WTPs) do not effectively remove PPCPs. Due to their extensive use, PPCPs are being detected extensively in the environment [3,7]. Figure 1 illustrates that the failure of WTPs to remove PPCPs means that the environment is constantly being polluted to the extent that they start to accumulate, becoming persistent and virtually irremovable organic pollutants. 

PPCPs occur in natural water bodies and wastewaters in minute concentrations, ranging from nanogram to microgram per litre (ng/L-µg/L), and are, hence, referred to as trace organics [1,8,9,10,11,12]. In recently reviewing the concentrations reported throughout the world, Adeleyae et al. [1] presented data for some wastewaters, where extremely high concentrations of PPCPs, up to 1 mg/L, were measured. Even though wastewater treatment plants (WWTPs) significantly remove most contaminants and many PPCP constituents, they remain the major sources of PPCPs in water bodies. Other primary pathways of entry into the environment are also shown in Figure 1 and discussed in the literature [2,3,7,13]. 

PPCPs do not have to occur in high concentrations in the environment to affect the ecosystem or quality of freshwater required for drinking purposes. They can damage human and animal health because their residues can eventually enter and accumulate in the food chain through effluent discharge and the reuse of treated sewage and sludge for agricultural applications [7]. PPCPs are widely distributed in the environment, which increases the risk of direct human exposure through the consumption of water and indirectly through the consumption of food, potentially compromising human health [3]. In reviewing the potential toxicity of PPCPs in water, Cizmas et al. [14] reported that human and animal exposures to PPCPs have linked them to an array of carcinogenic, mutagenic, and reproductive toxicity risks. Many PPCPs are reported to be toxic to freshwater invertebrates (such as daphniids), fish, mussels, and human embryonic cells at trace concentrations [15]. The types of toxic effects and test organisms used in the experiments and the concentrations of PPCPs at which the risks were noticed were presented by Pal et al. [15]. PPCP concentrations in some wastewaters were observed to be higher than the toxicity limits for some PPCPs [6,15,16]. 

Fick et al. [17] calculated the predicted critical environmental concentrations (CECs), i.e., the water concentration expected to cause a pharmacological issue in fish, for 500 PPCPs, assuming equivalent pharmacological activity. The CECs were derived from data available in the literature on their impact on humans and how each drug bio-concentrates in fish based on lipophilicity. The CEC values [17] were derived from toxicity studies that used a single compound and a single organism. This could mean that aquatic organisms can be chronically exposed to a combination of PPCPs and sometimes toxic at concentrations below the CECs of individual PPCPs. Many investigations regarding risk assessment have only considered the ecotoxicity of the parent drug, with very little attention paid to the additional contributions made by the metabolites of parent PPCPs excreted by microbes, humans and animals [18]. The combined effects of the parent PPCPs and their metabolites of several persistent PPCPs in water can be much greater than that of a single PPCP. Synergistic effects have been reported with PPCP mixtures, for example, ibuprofen and diclofenac, in the Daphnia test [19], but these effects may not be the same for all organisms [14].

Adeleye et al. [1] compared global PPCP concentrations in WWTP influent and effluent and in freshwater, with toxicity values determined in laboratory experiments. This comparison demonstrated that mortality data (LC50 and EC50) were much higher than PPCP concentrations entering and leaving the WWTP and entering the natural environment. However, the concentrations of some PPCPs in wastewaters were in the same range as sub-lethal concentrations. As stated before, it is possible that combined and continuous exposure to PPCPs can cause greater toxicity to aquatic organisms. PPCP concentrations in treated WWTP effluents and other water bodies were, on average, one to six orders of magnitude lower than the reported LC50 concentrations for fish, invertebrates, amphibians, algae and rotifers. 

The main purpose of WWTPs is to remove nutrients, pathogens and particulate matter from industrial and municipal wastewater; they are not designed to remove PPCPs. However, the concentrations of PPCPs decrease to different levels during their operation, depending on the characteristics of the PPCPs and the treatment conditions [1]. Adeleye et al. [1] examined from the literature the concentrations of six major classes of PPCPs at different sampling points in WWTPs, reported in the literature from selected countries, and noticed that, in general, the traditional primary treatment using clarification/sedimentation had low PPCP removal efficiency. The higher removal of PPCPs occurred during secondary treatment due to microbial degradation and adsorption to biomass. The polishing treatment, including chlorination, ozonation, advanced oxidation, adsorption or membrane filtration, further increased the removal efficiency and generally produced quality treated water. The efficiency of these polishing treatments can be significantly improved by combining some of these treatments.

The prevalence of PPCPs in wastewater and, more widely, in the environment have been reviewed in several articles; most can be grouped according to geographical focus [20,21], environmental media [22,23,24], type of drug [25,26] and toxic response [27,28]. Many reviews have discussed a variety of processes for removing PPCPs from the environment, especially water. These processes include adsorption [3,29,30,31], ozone-based processes [32,33], ozonation [16,34], membrane filtration [35,36], Fenton oxidation [16,37], UV oxidation [16] and biological processes [16,38]. However, these processes were not integrated into most studies to take advantage of the different mechanisms specific to each process and to be synergistic enough in the removal of the PPCPs. 

PPCP removal depends on their chemical structure, molecular weight, charge, hydrophobicity/hydrophilicity, hydrogen bonding, oxidising capability, etc. These properties vary widely among PPCPs, which explains the differences observed in the removal efficiency among PPCPs. For example, among PPCPs, the molecular weights vary from 89 to 791 [6], log Kow (octonol/water partition coefficient, a measure of hydrophobicity) varies from -2.1 to 13.9 [6] and electric charge can be positive, negative or neutral [36]. The efficiency of adsorption largely depends on the hydrophobicity and charge of the PPCP. For membrane filtration, it depends on molecular weight and charge. Additionally, the removal efficiency in all processes depends on the chemical structure of the PPCP. The amount removed varies according to the processes, particularly mechanisms of reactions with the PPCPs. Since there are varieties of PPCPs with different chemical and physical properties, one type of treatment may not remove all PPCPs. Combinations of treatments are necessary to remove the different types of PPCPs. Therefore, in this report, a brief review of the various individual treatment processes used in removing PPCPs, such as adsorption, oxidation, membrane filtration, and biological methods, is made, presenting their strengths and drawbacks. This is followed by combinations of processes, comparing them, where possible, with individual processes. In the review articles published thus far, very little information is available on the use of such integrated methods for the removal of PPCPs. This report attempts to fill this gap in knowledge.

## 2. PPCP Removal Methods

In general, treatment methods to remove PPCPs fall into three categories: biological, physical, and chemical. The activated sludge process, involving microbial degradation, is the main biological process and a secondary treatment process in WWTPs. Of the physical processes, adsorption and membrane separation are the most used ones. Among chemical methods, advanced oxidation processes (AOPs) have been widely employed in tertiary treatments [1]. These methods are discussed in this order in the following sections.

### 2.1. Conventional Biological Treatment Process 

Currently, organic contaminants are removed mainly by microbial degradation because of the low cost and unsophisticated operational requirements. Microorganisms remove organic contaminants either individually or together in a group by metabolic consumption. Additionally, they detoxify the contaminants by degradation with the enzymes they excrete. A detailed discussion on the pure and mixed cultures that degrade PPCPs are given elsewhere [15].

Activated sludge treatment is typically used in conventional WWTPs to remove PPCPs. Their removal using this process depends on the size and layout of the sludge bioreactor and treatment factors such as pH and retention times (both sludge and hydraulic) and is specific to the various types of PPCP compounds [37,38]. Appendix A demonstrates the extent to which a range of PPCPs is removed by WWTPs. Carbamazepine features prominently among the PPCPs that commonly occur in wastewater effluent [38]. This is not surprising since it is not readily removed by these processes, although the rate of removal depends on the actual nature of the activated sludge (e.g., microbial composition) and type of wastewater [37,39,40] Other PPCPs (ibuprofen, naproxen, ketoprofen, diclofenac, bezafibrate, sulfamethoxazole, trimethoprim) are removed in varying amounts, ranging from 40–100% [41,42,43,44].

The wide variability in the removal efficiency of each species of PPCP in the above- mentioned studies indicates the extent to which their removal is influenced by the environmental conditions of WWTPs, including the total and type of NOM in the water and seasonal variation. Kosma et al. [45] showed how at one WWTP in Greece, some PPCPs (paracetamol, bezafibrate, ibuprofen) were more effectively removed in summer rather than winter. Quednow and Püttmann [46] reasoned that this was due to the nature and amount of inflow affected by seasonal consumption and prevailing rainfall. Ma et al. [47] added that seasonal temperature variation was a clear influence on how biodegradation slows down in winter. 

Generally, the biodegradation of PPCPs is slower than their removal by physical and chemical methods. It takes days for biodegradation to remove most PPCPs. Some PPCPs can be toxic to microbes, and this can result in insufficient treatment. Additionally, some PPCPs have a high affinity to sludge, and this reduces the degradation ability of the microbes. For the reasons stated above, PPCPs are not sufficiently removed by conventional WWTPs even though their concentrations in the effluent are small. WWTPs’ effluents, which are typically discharged to rivers, can result in influent containing PPCPs in drinking water treatment plants (DWTPs), thus compromising drinking water security and human health. It is, therefore, imperative to remove PPCPs at WWTPs using advanced treatment processes (e.g., adsorption, ozone, Fenton oxidation, etc.). These processes and their effectiveness in removing a range of PPCPs are discussed next. 

### 2.2. Adsorption

#### 2.2.1. Activated Carbon (AC)

The adsorption process is often used for the successful removal of trace organic pollutants in water due to its simplicity, cost-effectiveness, efficiency at low concentrations, and minimal waste production [48]. Of the various adsorbents commonly available, AC is considered attractive for removing PPCPs from wastewater and is widely used not only in laboratory studies but also in pilot plant studies and full treatment plants [48,49,50,51]. Liu et al. [52] reviewed the effectiveness of PPCP removal by AC. Appendix A summarises the removal efficiency of PPCPs by AC reported in several studies (Wang and Wang [6]). There are several mechanisms for removing PPCPs by AC, which are presented in Figure 2 [53]. Of these, the hydrophobicity and charge interactions of AC and PPCPs are the main ones [3,11,16,36,48,53,54]. Jamil et al. [48] classified 17 PPCPs found in a reverse osmosis concentrate collected from a water reclamation plant in Sydney into four groups based on hydrophobicity (log Kow values) and charge. They showed that PPCP removal by adsorption on granular AC (GAC) was related to charge and hydrophobicity (Figure 3). The PPCPs that had a positive charge and high hydrophobicity values (log Kow > 3.5) had the highest removal rates. Rodriguez et al. [55] agreed that the adsorption capacity of AC depends on the hydrophobicity of the investigated PPCPs (3-methylindole, chloroprene and nortriptyline). PPCPs can be adsorbed by both GAC and powder AC (PAC). Meinel et al. [54] discovered that the latter was more effective in the removal of PPCPs. AC removal of PPCPs can be improved by using ideal operating conditions, for instance, contact time, etc. According to Wang and Wang [6], this is best done with pilot-scale studies. 

#### 2.2.2. Graphene

The structure of graphene comprises a single-layer two-dimensional array of carbon atoms arranged in a hexagon form that appears as a honeycomb sheet. Graphene is derived from graphene oxide, which, in turn, is produced when graphite is oxidised. Both the former materials have higher specific surface areas than AC [56,57] and, therefore, are expected to have greater adsorption capacities. However, the nanosheets of these materials can aggregate heavily in water due to π–π interactions and strong Van der Waals interactions between the graphene layers, which inhibit the materials’ high adsorption capacity. One promising strategy to overcome this problem is to incorporate graphene and graphene oxide nanosheets onto low-cost substrates [58]. They have been used in laboratory adsorption studies of PPCPs. These were predominantly batch experiments using synthetic wastewater and very high concentrations of PPCPs compared to actual wastewater (Appendix A), and therefore, the results may not be directly applicable to actual practice. Using very high PPCP concentrations would naturally produce higher adsorption capacities, and such high values are those that were reported in these studies. PPCP adsorption by graphene and graphene oxide should be investigated in pilot plants and full-scale plants using actual wastewater.

#### 2.2.3. Carbon Nanotubes (CNT)

The use of CNT for the adsorption of PPCP constituents such as ketoprofen, carbamazepine, sulfamethoxazole and triclosan has been studied [6,59,60,61,62]. These studies revealed that CNT is highly effective in removing PPCP constituents, such as those mentioned. The effectiveness is influenced by the CNT’s surface chemistry and properties, together with the PPCP’s physicochemical properties. Detailed discussions on these properties are provided by Wang and Wang [6]. 

While these forms of carbon adsorption (CNT, AC, graphene, graphene oxide) processes are promising, their application in large-scale scenarios is hampered by the following:(1)The costs of graphene and graphene oxide are still prohibitive, and further research is required to lower them [58];(2)The aggregation of graphene sheets should be prevented to avoid a reduction in adsorption capacity by loading it onto low-cost materials [58];(3)Similarly, more research is required to simplify and lower the cost of CNT production;

As part of large-scale applications, it is important to regenerate the various forms of carbon adsorbents (AC, CNT, etc.), once they become exhausted, for reuse/recycling purposes [6]. 

### 2.3. Membrane Process

Water treatment plants (WTPs) now employ a variety of membrane processes to remove pollutants: reverse osmosis (RO), nanofiltration (NF), ultrafiltration (UF), and microfiltration (MF). Membranes used in MF, UF and NF are distinguished by the size of their pores, which are about 0.1, 0.01 and 0.001 µm, respectively [62]. The larger the pore size, the lower the trans-membrane pressure (TMP) required, translating to lower energy and operating costs. However, membranes of a larger pore size remove lower amounts of smaller-sized pollutants.

Membranes remove PPCPs through the processes of size exclusion, electrostatic repulsion, and adsorption [35,63,64]. The removal efficiency depends on numerous factors grouped as the PPCPs’ physicochemical (size, charge, hydrophobicity) and membrane properties (pore size, molecular weight cut-off (MWCO), zeta potential, membrane-solute interactions) [35]. The removal of PPCPs can, therefore, be complex, and its efficiency relies on numerous factors.

MF and UF membranes, because of their large pore size, are generally used to remove suspended solids. PPCPs are not removed because their molecular weights (MWs) are typically in the 200–800 Da range (or approximately 0.000025–0.0001 micrometres), while MF and UF membranes’ MWCO are several thousand Daltons [63]. PPCPs can be removed by NF and RO processes, as demonstrated by Couto et al. [35]. RO and NF membranes, which are commercially available, vary in charge, MWCO and hydrophobicity/hydrophilicity, which influence the efficient removal of specific PPCPs. The PPCPs themselves exist in a range of properties, so it is not possible to generalise their performance in terms of one property. Couto et al. [35] and Taheran et al. [63] have summarised the PPCP removal performance of various types of RO and NF membranes, although many of the studies summarised were conducted using synthetic and ultra-pure water. Only a few studies, for example, those conducted by Urtiaga et al. [65], were done with secondary-treated wastewater.

Urtiaga et al. [65] conducted a long-term combined UF and RO pilot plant study in northern Spain, treating raw municipal wastewater and secondary-treated effluent. In this study, 12 PPCPs (caffeine, nicotine, naproxen, ibuprofen, ofloxacin, furosemide, hydrochlorothiazide, gemfibrozil, bezafibrate, fenofibric acid, atenolol and N-acetyl-4-amino-antipyrine (4-AAA)) were monitored. Removal efficiency for most PPCPs that were monitored was quite low (less than 20%) when only UF was used. Where UF was followed by RO, more than 99% removal was achieved. The system was operated at a low TMP of 11 bars, resulting in low energy consumption.

Jamil et al. [36] used MF-treated water from a WTP in Sydney, Australia, that treated domestic sewage and stormwater in a study of the NF removal of PPCPs; 10 types of PPCPs with molecular weights between 119–296 g/mol and negative or neutral charge were monitored. An NF90 membrane was used (membrane MWCO = 90–200 Da, moderately hydrophobic, negatively charged). The results of this laboratory study, summarised in Table 1, show that the removal rate was between 35% to >98%. According to Jamil et al. [36], the large variation in removal efficiency was caused by the PPCPs’ charge, molecular weight and degree of hydrophobicity differences.

Seven of the ten PPCPs monitored by Jamil et al. [36] were removed by >90% without any pre-treatment (NF alone). The NF membrane was negatively charged and could achieve >90% removal by the electrostatic repulsion of four negatively charged PPCPs (diclofenac, gemfibrozil, ibuprofen, and naproxen), no matter what their molecular weights were. Of these, the size exclusion mechanism would have also excluded diclofenac, the PPCP with the largest MW (296 g/mol). NF90 excluded triclosan and trimethoprim, which were both neutral in charge and likely excluded by size since the membrane’s MWCO was smaller than the PPCPs’ MW (290 g/mol). Triclosan was also removed, in fact, by >90%. It is highly hydrophobic (log Kow 4.76) and could have been removed by adsorption onto the membrane, which was moderately hydrophobic. 

PPCPs could also have been removed by adsorption onto organic materials in the feed that was deposited on the membrane during the NF process. Saccharin and benzotriazole were not removed at high rates (88% and 35% rejection, respectively). Both had a neutral charge and the lowest MW (183 and 119 g/mol, respectively). The low rate of removal was because these PPCPs were able to pass through some of the larger membrane pores. Diuron, with an MW of 233, within the range of the membrane’s MWCO, also had a low removal rate (77%) because some of the PPCP’s molecules could have passed through the larger pores of the membrane. 

A drawback of the membrane processes is fouling (due to material deposition and/or biofilm formation on the membrane), requiring chemical cleaning [66]. The frequency of cleaning can be significantly reduced by applying pre-adsorption to remove foulants or by using a submerged membrane adsorption hybrid system (SMAHS) to remove PPCPs. The adsorbents used in the SMAHS study were GAC and purolite. Another problem with the membrane process is the concentrate produced in the process, which is 3–5 times more concentrated with PPCPs and other contaminants. The improper discharge of this concentrate may trigger much greater potential health risks to non-target species, particularly those in aquatic environments, than the original wastewater. Therefore, this concentrate needs to be adequately treated before discharge to water bodies. 

### 2.4. Advanced Oxidation Processes

PPCPs in wastewater are not normally removed by typical wastewater treatment processes and require advanced treatment methods for removal. Toxic organic pollutants in wastewater, which are recalcitrant (i.e., PPCPs), may be removed by advanced oxidation processes (AOPs). These include ozonation, O_3_/UV, UV/H_2_O_2_, Fenton and Fenton-like oxidation, gamma radiolysis, sonolysis and electrochemical oxidation [67]. During these processes, by-products may form. They need to be removed by processes such as adsorption [52], or else they can hinder the oxidation process.

#### 2.4.1. Ozonation

Ozonation is the most commonly used method to remove PPCPs and is the most studied oxidation process. During ozonation, hydroxyl free radicals (OH•) are formed. The ozonation rate of PPCPs depends on the concentration of OH• and ozone. OH• oxidation potential (2.8 V) is higher than ozone (2.07 V). To develop more OH•, hydrogen peroxide is used to decompose ozone in solution. Ozonation is usually used as a post-treatment process and has been shown to remove most PPCPs. Removal of PPCPs by ozonation has been reviewed by Esprugas et al. [68]. 

In ozonation, the PPCPs’ removal rates depend on the ozone kinetic rate constant (kO_3_) and the OH• kinetic rate constant (k OH•) [34,69]. The ozonation increases with kO_3_ and k• OH based on the results of a pilot plant treatment study of secondary effluent (0.7 mg O_3_/mg DOC) [33]. The PPCPs were classified into fast (>90% removal), moderate (40–80% removal) and slow (0–70% removal) reactivity with ozone (Figure 4).

The DOC in wastewater can vary in concentration and type. In oxidation, DOC can compete with PPCPs for OH•, and in this way, the ozonation of PPCPs can wane significantly. The range of pollutants in wastewater is large and depends on its source. More research is required to study the effectiveness of oxidation of PPCPs in the presence of other pollutants that compete for OH• in different types of wastewater. The reduction in the ozonation rate of PPCPs depends on the ozone rate constants of the individual DOC fractions [70]. Another area of concern is the fate of ozonation by-products, which may be toxic in the environment. 

#### 2.4.2. Fenton Oxidation 

In Fenton oxidation, highly reactive free radicals (hydroxide radical (•OH), sulphate radical (SO_4_ •−) and superoxide radical (O_2_ •−)) are produced in-situ. These radicals have a strong ability to oxidise PPCPs. They form with the release of precursor oxidants, for example, hydrogen peroxide, persulfate/peroxodisulfate, peroxymonosulfate and sodium percarbonate [37]. These precursor oxidants are released utilising a variety of methods, such as metal-based catalysts (Fe, Mn, Co, Cu, V, Ru, Mo, Cr, Ce), heat, UV radiation or visible light, ultrasound, alkaline aqueous medium, etc. 

The chemical equation for the catalytic decomposition of hydrogen peroxide by the reaction with iron salts to produce hydroxide radicals is shown in Equation (1) [37].
Fe^2+^ + H_2_O_2_ + H^+^ → Fe^3+^ + H_2_O + •OH (1)

In comparison to hydroxide radicals (redox potential 2.8 V), the sulphate radicals (2.5–3.1 V) produced from a persulphate possess even higher redox potential and can degrade PPCPs to a similar or better capacity [16]. In the thermal activation of persulphate, hydroxyl is the main radical produced (Equations (2)–(4)). However, in activation under alkaline conditions, sulphate and superoxide are the main radicals produced (Equations (5)–(6)).
S_2_O_8_ ^2−^ → 2 SO_4_ •− (2)
HSO_5_ − → SO_4_ •− + •OH (3)
SO_4_ •− + H_2_O → SO_4_ ^2−^ + •OH + H^+^
(4)
S_2_O_8_ ^2−^ + H_2_O → 2SO_4_ ^2−^ + HO_2_^−^+ H^+^
(5)
S_2_O_8_ ^2−^ + HO_2_^−^ → SO_4_ ^2−^ + SO_4_ •− + O_2_ •− (6)

Several limitations prevent the Fenton process from being applied on a large scale. These include the narrow pH range required, the precipitation of metals in the catalyst in some types of actual wastewater that leads to sludge formation, and the formation of toxic by-products. These may be overcome partly by using heterogeneous catalysts. Notwithstanding the efficiency of homogeneous systems (liquid phase alone), the additional expense of iron salt removal from the environment is too costly. As a result, heterogeneous catalysts (solid–liquid phases) have been sought to improve catalysis. 

An important Fenton oxidation process uses iron salts and hydrogen peroxide in an acid environment to treat industrial wastewater. Fenton oxidation, similar to ozone oxidation, relies on the OH• oxidising capacity. In recent times, other types of Fenton processes have been formulated, for instance, electro-Fenton and photo-Fenton oxidation. The former has been reviewed by Feng et al. [71], while Fenton-like systems have been explored by Bokare and Choi [72]. Both give detailed information on the mechanisms for PPCP removal through Fenton oxidation and Fenton-like systems. 

Appendix A summarise the information on the conditions and results for the Fenton oxidation of PPCPs, as sourced from previous studies. However, in these studies, synthetic waters/pure waters were implemented with very high PPCP concentrations, relative to that in municipal wastewaters. Nonetheless, the effective removal of PPCPs using Fenton oxidation or Fenton-like oxidation was demonstrated. In these processes, H_2_O_2_ is decomposed by various metal-based catalysts to generate •OH radicals. The solubility of metal-based catalysts limits homogenous catalysis (liquid phase). Other highly soluble metal-based catalysts (e.g., cerium, cobalt, etc.) are not used because they are cytotoxic. Heterogeneous catalysis (solid–liquid phase), by comparison, are not similarly affected. Nevertheless, the unstable nature of heterogeneous catalysts and their recycling potential need to be addressed. It is, therefore, necessary to resolve these issues for Fenton-like systems [73]. The environmental fate of the by-products produced needs to be studied, and typically, other treatment processes will be required to remove them. Studies should also be conducted in a mix of PPCPs and NOM, the latter affecting the removal of a particular PPCP.

#### 2.4.3. UV Oxidation Treatment 

A popular water treatment method to disinfect water for potable purposes is ultraviolet (UV) treatment. Similarly, UV is used to disinfect wastewater effluent that has undergone biological treatment and sand filtration processes for reclaimed water applications where there is potential direct contact. PPCP removal is also possible with UV [74]. In photolysis, UV breaks the chemical bonds of PPCP constituents and removes them. However, some constituents, such as carbamazepine, are not significantly affected by UV photolysis and are not effectively removed [6,75]. To better treat PPCPs, in the advanced oxidation process (AOP), hydrogen peroxide is coupled with UV (Appendix A). This process has proven to be effective in removing PPCPs [76]. 

#### 2.4.4. Electrochemical Advanced Oxidation Processes (EAOPs) 

Electrochemical advanced oxidation processes (EAOPs) have garnered increasing attention during the last few decades as an attractive group of AOPs [77,78]. It involves anodic oxidation, where organics can be directly oxidised at the anode surface by electron transfer and/or indirectly oxidised by •OH weakly adsorbed at the anode surface and/or agents in the bulk solution, such as active chlorine species, O_3_, persulfates and H_2_O_2_ [78]. These processes have been successfully applied to remove various pollutants from wastewaters [77,78], including PPCPs [79].

Lozano et al. [79] reviewed the removal of various PPCPs using EAOPs, namely, anodic oxidation (AO), electro-Fenton, photoelectron-Fenton, solar photoelectron-Fenton, photo-electrocatalysis and sono-electrochemical processes. They reported that AO is one of the most straightforward methods for degrading organic compounds, and it has successfully removed a large percentage of several PPCPs. However, applying EAOPs can be expensive due to high electrode and operational costs, which include electrical energy for electrochemical cells and plant operation, reagents and maintenance. The studies conducted so far have used much higher concentrations of PPCPs than those quantified in wastewater. Additionally, they used synthetic water rather than real wastewater. Consequently, studies on actual wastewater need to be undertaken.

### 2.5. Removal of PPCPs by Combined Methods

Existing individual water treatment processes can either degrade or remove pharmaceuticals. PPCP removal is commonly achieved by phase transfer methods such as sorption or membrane filtration. These methods do not lead to pollution removal/reduction as they generate a concentrated phase in addition to treated water. The way to degrade pharmaceuticals is via advanced oxidation processes. However, those cannot completely mineralise persistent compounds at low concentrations and lead to oxidation by-products with toxicity [80]. Thus, the existing removal and degradation methods are not solutions for PPCP removal in the environment. Therefore, this review focuses on combined treatment processes, as discussed below.

#### 2.5.1. Combined Chemical and Biological Methods

Biological treatment has proven to be ineffective at removing persistent pollutants such as PPCPs because they are toxic to microorganisms or can resist their activities. AOPs, however, are effective in removing these pollutants. In this process, intermediates that are not easily oxidised may be produced. This prolongs treatment times, consuming more energy and increasing costs. Here, a combined AOP and biological treatment strategy can be effective [81]. AOP serves as a pre-treatment process for persistent pollutants. The intermediates that form can be biologically treated, degraded and removed completely [6]. 

PPCPs are effectively removed by the combined process of AOP/biological treatment. De Wilt et al. [82] developed a three-stage process comprising biological treatment, followed by ozonisation, and then biological treatment for the removal of PPCPs (caffeine, carbamazepine, diclofenac, gemfibrozil, ibuprofen, metoprolol, naproxen, sulfamethoxazole and trimethoprim) from a secondary-clarified effluent sourced from a WWTP in the Netherlands, spiked with known concentrations of PPCPs. This process proved cost-effective. The first biological treatment step removed 38% of ozone-scavenging TOC, thus proportionally reducing the absolute ozone input (dose) for the removal of biorecalcitrant PPCPs in the subsequent ozone treatment. The second biological treatment removed the potentially toxic by-products formed during ozonation.

#### 2.5.2. Combined Chemical and Physical Methods (Ozonation and Adsorption)

While ozonation can deactivate microorganisms, NOM and PPCPs, oxidation by-products can form. Some PPCPs are slowly oxidised and cannot be practically removed [34,69]. At the ozone doses used in conventional wastewater treatment, NOM and PPCPs compete for oxidants, leaving a portion of the latter unoxidised [70]. For these reasons, a post-treatment of adsorption is added after ozonation to treat the remaining NOM, PPCPs and their by-products. Many investigations have used AC for this purpose [34,83,84,85]. 

Zietzschmann et al. [83] investigated the ozonation process at various ozone dosages for the removal of NOM and PPCPs contained in wastewater effluent spiked with 12 PPCPs. A post-treatment of PAC adsorption followed to remove oxidised by-products and any remaining unoxidised NOM and PPCPs. Ozonation followed by PAC adsorption post-treatment reduced PPCP concentrations more effectively than individual treatments. Ozonation reduced the competition for adsorption by NOM constituents. This was because NOM constituents were transformed into compounds with poorer adsorption capacity. This is attributed to their lower aromaticity, molecular size, and hydrophobicity. Other studies showing the advantage of combining ozonation and adsorption processes have been reported in a recent review paper [53]. Additional information is given in Appendix A.

#### 2.5.3. Combined Membrane Processes

The combination of adsorption with the MF or UF process is a simple and cost-effective treatment strategy. It combines the advantage of adsorption with membrane filtration’s effectiveness in particle removal. Of the adsorbents, AC was found to be popular, and it has the additional advantage of being able to biodegrade PPCPs. Although AC has been used before or after membrane filtration, the former is more popular. A recent study [36] using microfiltered wastewater investigated the removal of 10 PPCPs using NF, either alone or after pre-treatment with two adsorbents (GAC, Purolite ion exchange resin). These adsorbents contained different properties that influenced PPCP removal. It was found that >90% of seven PPCPs were removed by NF without pre-treatment (Table 1). However, the remaining three PPCPs required an adsorption pre-treatment for satisfactory removal. Benzotriazole removal was 35% with NF alone. Removal with the GAC+NF process was 94%, and with the Purolite+NF process, it was 99%. The corresponding removals for diuron were 77%, >94% and >94%, respectively. Saccharin removals were 88%, >92% and >92%, respectively.

A more efficient method than using membrane filtration and adsorption in the sequence is combining both processes together in a single tank containing the influent, where the membrane is submerged and adsorbent suspended. This process is called SMAHS, and its success is mainly due to the following [86]:

Membrane anti-fouling: An air diffuser placed at the bottom of the influent tank creates coarse air bubbles that are used to keep the adsorbent in suspension. The bubbles also flow past the membrane surface, inducing shear stress across it and removing the membrane foulant. The energy requirement of immersed membrane systems used in wastewater treatment plants is currently very low (less than 20% of the total energy requirement). Optimisation of backwash: For the successful long-term operation of the membrane process, it is necessary to optimise the frequency and duration of the backwash. Adaptive backwash initiation and duration schemes with new control systems can lead to a 40–50% reduction in backwash water and energy consumption [87,88]. Incorporation of adsorbent in SMAHS: The adsorbent added to the SMAHS creates an additional shearing effect that reduces particle deposition on the membrane surface and reduces membrane resistance. It directly removes organics that would otherwise deposit on the membrane and cause fouling. The periodic daily substitution of adsorbent is as little as 2–5%, which is equivalent to an average adsorbent residence time of 20–50 days in the tank. This helps economise the adsorbent without it becoming exhausted.

#### 2.5.4. Advanced Membrane Bioreactor Hybrid Systems

Electrodialysis [89] and forward osmosis [90] can be used to remove organic micropollutants (OMPs, mostly PPCPs) from wastewater. However, in this paper, we consider only the combined advanced membrane bioreactor hybrid systems. Membrane bioreactors (MBRs), which combine biological reactors and membrane separation, are a promising option in wastewater treatment as they generate clean effluent. The effluent is almost free of suspended solids, microorganisms and OMPs. The footprint is smaller, with lower sludge disposal costs compared with conventional biological treatment. Hydrophilicity and hydrophobicity are important aspects of OMP removal. The hydrophobicity of an organic molecule is defined by the octanol-water partitioning coefficient (Kow) or the solid water partitioning coefficient (Kd) [91]. Compared to negatively charged or neutral OMPs, the positively charged pharmaceutical class OMPs showed more affinity towards sludge adsorption in MBRs [91]. 

A hybrid MBR can produce better-quality effluent with low membrane fouling. This, in turn, reduces cleaning frequency [92]. Appendix A compares conventional MBRs with two major advanced hybrid MBR systems, namely, osmotic membrane bioreactors (OMBRs) and membrane distillation bioreactors (MDBRs), for their effectiveness in wastewater treatment. A detailed comparison can be found in Pathak et al. [93]. 

##### Osmotic Membrane Bioreactor

Osmotic membrane bioreactors (OMBRs) are employed in wastewater treatment systems to reclaim and reuse indirect and direct potable water sources [93] by integrating semi-permeable forward osmosis membranes with a bioreactor. OMBRs achieve better permeate quality, with lower dissolved organic matter, lower fouling tendency, higher reversibility of membrane fouling and the improved removal of organic micropollutants [94]. 

Appendix A presents some recently published OMBR studies on OMP removal. A more detailed list can be found in Pathak et al. [93].

##### Membrane distillation bioreactor

In this system, membrane distillation incorporates a hydrophobic microporous membrane operating at a low temperature, which solely involves the transfer of water vapour from the feed side to the distillate side through membrane pores. Due to gas-phase mass transfer, only volatile matter may pass through, and thus, MD completely retains non-volatile matter in the feed solution [95]. The membrane distillation bioreactor (MDBR) has been studied for its ability to integrate membrane distillation and conventional biological systems in a single reactor. The direct contact membrane module is submerged into the activated sludge tank. 

Wijekoon et al. [95] evaluated the performance of MDBRs in OMP removal and concluded that 95% of OMPs can be removed by this process; biodegradation contributed to 70% of OMP removal. Appendix A presents some recently published MDBR studies for OMP removal. A detailed list can be found in Pathak et al. [93].

##### Comparison of Hybrid MBRs

The life cycle assessment (LCA) is a significant tool to measure the environmental impact of different wastewater treatment schemes in order to compare their performances in terms of energy and greenhouse emissions and cost components [96]. 

UF-OMBR or (FO-MBR) has the potential to become a fourth-generation advanced wastewater reclamation alternative, providing FO membrane development and OMBR process optimisation are accomplished [97]. The UF-OMBR consists of UF and FO membranes in a bioreactor. UF produces non-potable reuse water, while FO produces potable quality water.

## 3. Concluding Remarks and Perspectives

Biological wastewater treatment processes, in general, and the activated sludge process, in particular, are ineffective treatments for PPCP removal. Based on the composition of PPCPs and their chemical characteristics (molecular weight, charge, hydrophobicity/hydrophilicity, functional groups, chemical structure), different processes need to be employed for their effective removal. Other processes, such as AOP, are useful pre-treatments that decompose PPCPs into intermediates that can be removed by biological treatment. 

Ozonation, a common AOP, followed by adsorption post-treatment, can be successful in removing any unoxidised PPCPs and their oxidation by-products. The competition between NOM and PPCPs in wastewater for oxidation and adsorption can influence PPCP removal. Ozone dose adjustment and appropriate adsorbent selection may be applied to overcome this. Lastly, the combination of adsorption with membrane filtration is an efficient method of removing PPCPs compared to using either process alone. It combines the advantage of adsorption and the ability of membrane filtration to effectively remove particles. The combined process is either applied sequentially, usually with adsorption before membrane filtration, or integrated into the SMAHS process, where the membrane is submerged and the adsorbent suspended in the tank containing the influent. The benefits of SMAHS are membrane de-fouling, greatly improved backwash, and prolonged membrane life. The integration of the processes takes advantage of the different mechanisms specific to each process and provides a synergistic effect on the removal of the PPCPs. OMBR or (FO-MBR) has the potential to become a fourth-generation advanced wastewater reclamation alternative, with ongoing and future membrane development and OMBR process optimisation.

## Figures and Tables

**Figure 1 membranes-13-00158-f001:**
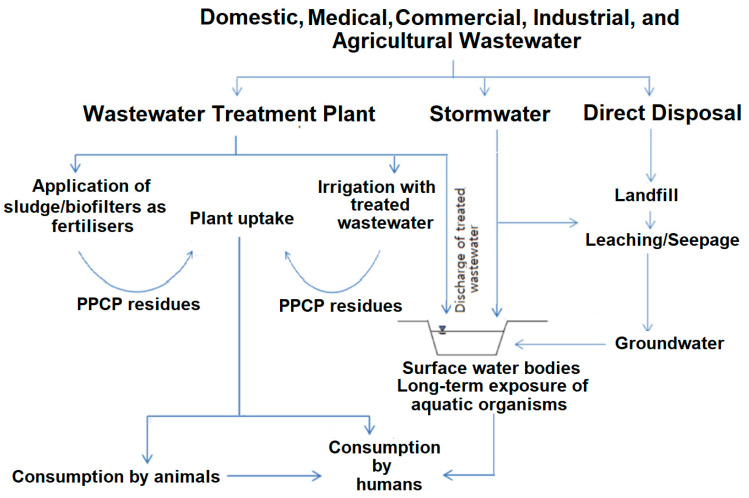
Sources and pathways of PPCP entry into the natural environment (modified from Reyes et al. [2]).

**Figure 2 membranes-13-00158-f002:**
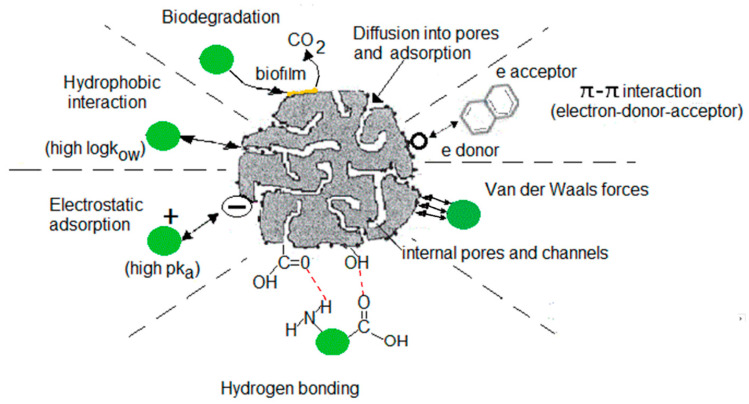
Possible mechanisms for removal of PPCPs from wastewater using the AC adsorption process (modified from Loganathan et al. [53]).

**Figure 3 membranes-13-00158-f003:**
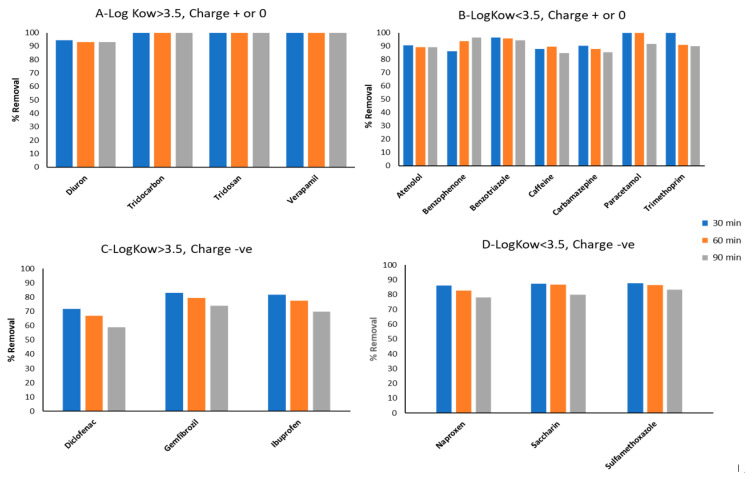
Classification of PPCPs into four groups based on hydrophobicity (Log Kow—higher values indicate greater hydrophobicity) and charge and their removal from ROC collected in a water reclamation plant by GAC columns in a laboratory study after different breakthrough times (modified from Jamil et al. [48]).

**Figure 4 membranes-13-00158-f004:**
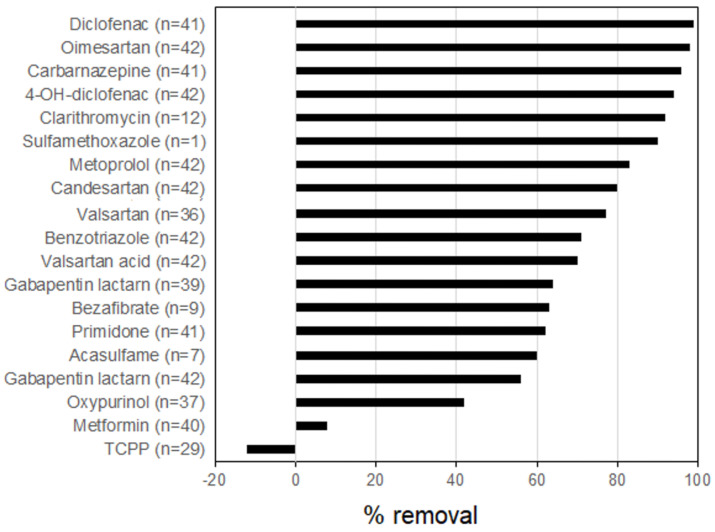
Average percentage removal of 19 monitored PPCPs during ozone treatment of wastewater in a pilot plant study ((n is the number of data points) (redrawn from Sauter et al., [34]).

**Table 1 membranes-13-00158-t001:** Removal of 10 PPCPs from MF wastewater by NF and NF with adsorption pre-treatment [36].

PPCP	By NF alone	By GAC + NF	By Purolite + NF
Benzotriazole	35	94	99
Carbamazepine	96	96	>98
Diclofenac	>93	>93	>93
Diuron	77	>94	>94
Gemfibrozil	>95	>95	>95
Ibuprofen	>90	>90	>90
Naproxen	>98	>98	>98
Saccharin	88	>92	>92
Triclosan	>92	>92	>92
Trimethoprim	>97	>97	>97

## Data Availability

Not applicable.

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
