# Peer review of "Treatment Trends and Combined Methods in Removing Pharmaceuticals and Personal Care Products from Wastewater—A Review"

_membranes, 2023, doi:10.3390/membranes13020158_

Round 1

Reviewer 1 Report

The manuscript reviewed the treatment trends and combined methods in removing pharmaceuticals and personal care products from wastewater. The PPCPs removal methods and improvements for removal strategies were discussed.  However, the membrane process discussed here might be inadequate. Only the references of reverse osmosis, nanofiltration, ultrafiltration, and microfiltration were reviewed. Other membranes technologies such as electrodialysis or emerging technologies were not included. The manuscript must be improved in this aspect.

Author Response

Reviewer 1

Q1.Only the references of reverse osmosis, nanofiltration, ultrafiltration, and microfiltration were reviewed. Other membranes technologies such as electrodialysis or emerging technologies were not included. The manuscript must be improved in this aspect.

Response: As suggested by the reviewer, other advanced membrane processes (more importantly, two major advanced hybrid MBR systems, namely (Forward osmosis) osmotic membrane bioreactors (OMBR) and membrane distillation bioreactors (MDBR)) are included in the revised manuscript (lines 519 to 566). Four new Tables S9 to S12 are also included to show their advantages and their PPCP  removal capacity. These revisions are highlighted in yellow.

Reviewer 2 Report

The manuscript discusssed different methods to remove micro pollutants from wastewater, the MS is readable and easy to understand, most of the information is a general knowledgable that can be found in books, but the authors were successful to correlate the known process with research results which gave the MS a gig her value. I would recommend publications because it helps researcher see in the demain to know about the others findings. A few comments to edit before publishing:

line 216 CNT was or were

line 235 please check the order UF, MF and NF

lines 246 to 248 can you add the corresponding pore size in Micrometer to the given Dalton unit

line 322 please check where the dot is written before or after OH, please check everywhere ( lines 338, 363 and 387 as examples) 

line 388 O3, subscript

line 433 " while ozonation can remove MO" Inthing the word remove is not suitable, I suggest to use deactivate.

line 436 " compete for oxidation " or compete for oxidants

line 470 membrane defouling or membrane anti- fouling measure

line 491 is or are?

in the conclusion part, I suggest to remove the numeration, firstly until fourthly. Remove only the nemeration bet keep the texts as it is .

Author Response

Reviewer 2

I would recommend publications because it helps researcher see in the demain to know about the others findings. A few comments to edit before publishing:

Comments: line 216 CNT was or were

line 235 please check the order UF, MF and NF

lines 246 to 248 can you add the corresponding pore size in Micrometer to the given Dalton unit

line 322 please check where the dot is written before or after OH, please check everywhere ( lines 338, 363 and 387 as examples)

line 388 O3, subscript

line 433 " while ozonation can remove MO" Inthing the word remove is not suitable, I suggest to use deactivate.

line 436 " compete for oxidation " or compete for oxidants

line 470 membrane defouling or membrane anti- fouling measure

line 491 is or are?

in the conclusion part, I suggest to remove the numeration, firstly until fourthly. Remove only the nemeration bet keep the texts as it is

Response:

All the above corrections are made in the revised manuscript and highlighted in yellow.

Reviewer 3 Report

Reviewer Comments

COMMENTS TO AUTHOR:

The review entitled “Treatment trends and combined methods in removing pharmaceuticals and personal care products from wastewater – a review” by Saravanamuthu and co authors comprehensively focus on the removing pharmaceuticals and personal care products from wastewater.

   I think it is of great interest in the community of waste water treatment. As a result, I will recommend the publication of this manuscript to accept it followed by some minor corrections.

Comments.

1.    Please check English in overall manuscript.

2.    Kindly expand the introduction section.

3.    Kindly expand Fenton oxidation section and include some chemical reactions for the same for understanding the process.

4.    Please make same format for all the references. Reference 1, 2 and 3 are in different formats.

Author Response

Reviewer 3

I think it is of great interest in the community of waste water treatment. As a result, I will recommend the publication of this manuscript to accept it followed by some minor corrections.

Comments.

1Please check English in overall manuscript.

Response: The  manuscript is checked by English editor.

  1. Kindly expand the introduction section.

Response: As per the advice of reviewer, the introduction section is expanded in the revised manuscript (please see  lines 96 to 108 and lines 118 to 126 as highlighted in yellow).

  1. Kindly expand Fenton oxidation section and include some chemical reactions for the same for understanding the process.

Response: As per the advice of reviewer, this  section is expanded in the revised manuscript with chemical reactions (please see  lines 358  to 377 as highlighted in yellow).

  1. Please make same format for all the references. Reference 1, 2 and 3 are in different formats.

Response: References are formatted.

Round 2

Reviewer 1 Report

The manuscript has been improved, and can be published on Membranes.